# Exploring the Neuroprotective Potential of *Desmodium* Species: Insights into Radical Scavenging Capacity and Mechanisms against 6-OHDA-Induced Neurotoxicity

**DOI:** 10.3390/plants13131742

**Published:** 2024-06-24

**Authors:** Hung-Chi Chang, Jin-Cherng Lien, Min-Chung Hsueh, Chi-Rei Wu

**Affiliations:** 1Department of Golden-Ager Industry Management, College of Management, Chaoyang University of Technology, Taichung 413310, Taiwan; changhungchi@cyut.edu.tw; 2School of Pharmacy, China Medical University, Taichung 40402, Taiwan; jclien@mail.cmu.edu.tw; 3Department of Chinese Pharmaceutical Sciences and Chinese Medicine Resources, China Medical University, Taichung 40402, Taiwan; rei202204@gmail.com

**Keywords:** *Desmodium* genus, *Desmodium pulchellum*, radical scavenging capacity, neuroprotective effects, 6-hydroxydopamine, neurotoxicity, Parkinson’s disease

## Abstract

In this study, we collected seven prevalent Taiwanese *Desmodium* plants, including three species with synonymous characteristics, in order to assess their antioxidant phytoconstituents and radical scavenging capacities. Additionally, we compared their inhibitory activities on monoamine oxidase (MAO) and 6-hydroxydopamine (6-OHDA) auto-oxidation. Subsequently, we evaluated the neuroprotective potential of *D. pulchellum* on 6-OHDA-induced nerve damage in SH-SY5Y cells and delved into the underlying neuroprotective mechanisms. Among the seven *Desmodium* species, *D. pulchellum* exhibited the most robust ABTS radical scavenging capacity and relative reducing power; correspondingly, it had the highest total phenolic and phenylpropanoid contents. Meanwhile, *D. motorium* showcased the best hydrogen peroxide scavenging capacity and, notably, *D. sequax* demonstrated remarkable prowess in DPPH radical and superoxide scavenging capacity, along with selective inhibitory activity against MAO-B. Of the aforementioned species, *D. pulchellum* emerged as the frontrunner in inhibiting 6-OHDA auto-oxidation and conferring neuroprotection against 6-OHDA-induced neuronal damage in the SH-SY5Y cells. Furthermore, *D. pulchellum* effectively mitigated the increase in intracellular ROS and MDA levels through restoring the activities of the intracellular antioxidant defense system. Therefore, we suggest that *D. pulchellum* possesses neuroprotective effects against 6-OHDA-induced neurotoxicity due to the radical scavenging capacity of its antioxidant phytoconstituents and its ability to restore intracellular antioxidant activities.

## 1. Introduction

Parkinson’s disease (PD) is a chronic neurodegenerative disorder affecting the central nervous system (CNS), typically afflicting individuals over the age of 55. Its primary clinical manifestations include motor impairments such as resting tremor, rigidity, bradykinesia, and postural instability, which stem from progressive neuronal degeneration in the nigrostriatal dopaminergic pathway. While remains a cornerstone therapeutic agent for PD, long-term treatment often leads to side effects such as levodopa-induced fluctuations and dyskinesias. In response, medical scientists have developed targeted drugs aimed at enhancing the activity of the central dopaminergic neuronal system, notably dopamine agonists, catechol-O-methyltransferase (COMT) inhibitors, and monoamine oxidase (MAO) inhibitors, particularly the latter [1]. MAO exists in two forms, MAO-A and MAO-B, each selectively distributed across various organs and preferentially metabolizing different neurotransmitters. MAO-A primarily metabolizes norepinephrine and serotonin, while MAO-B predominantly acts on dopamine. Consequently, their respective selective inhibitors have played pivotal roles in modulating neurotransmitter metabolism and have been utilized as therapeutic agents for conditions such as depression and PD [2]. Given that the oxidative deamination of dopamine catalyzed by MAO-B generates hydrogen peroxide and aldehydes, easily leading to oxidative damage in nerve cells, irreversible MAO-B inhibitors such as selegiline and rasagiline have demonstrated antioxidant and neuroprotective properties in in vitro and in vivo PD models. As a result, MAO-B inhibitors have been recognized for their multifaceted pharmacological activities and employed by clinicians in the management of PD [3]. Some researchers are actively exploring the potential of plant-derived or natural products with MAO-B inhibitory activity and neuroprotective effects as promising therapeutic agents for PD [4].

The causes of PD are multifactorial, encompassing genetic predisposition, infections, and exposure to environmental toxins. Epidemiological investigations have linked chronic exposure to herbicides and pesticides with an increased risk of PD. The induction of Parkinsonism by 1-methyl-4-phenyl-1,2,3,6-tetrahydropyridine (MPTP) strongly supports the environmental toxin theory of PD. Additionally, subsequent research has revealed that the herbicide paraquat and the pesticide rotenone can also trigger PD-like symptoms [5]. These effects of these environmental toxins elucidate several key neuropathological mechanisms underlying PD. Two major hypotheses have emerged regarding the pathogenesis of PD. One prominent theory implicates mitochondrial dysfunction and subsequent oxidative stress, particularly involving toxic oxidative dopamine (DA) species such as DA-quinone. The metabolite 1-methyl-4-phenylpyridium (MPP^+^) derived from the metabolism of MPTP by MAO-B or rotenone selectively inhibits mitochondrial complex I of the electron transport chain. This inhibition disrupts oxidative phosphorylation, impairs ATP production, and elevates the generation of reactive oxygen species (ROS). Under normal circumstances in dopaminergic neurons, dopamine auto-oxidation and dopamine metabolism are prone to produce large amounts of ROS, not to mention the presence of the above-mentioned environmental toxins. Excessive ROS production leads to aberrant protein processing and triggers the apoptosis of dopaminergic neurons. Consequently, targeting cellular redox homeostasis to mitigate intracellular oxidative stress represents a promising therapeutic approach for PD [6]. Building upon these insights into PD neuropathology, 6-hydroxxydoppamine (6-OHDA) and rotenone are commonly employed as tools to establish in vitro and in vivo models replicating PD-relevant neuropathology and behavioral phenotypes. These models facilitate the assessment of the potential efficacy of plants, natural products, or compounds as therapeutic interventions for the prevention and treatment of PD [7].

The genus *Desmodium*, a member of the Fabaceae family, has a long history of use as traditional folk medicine in Taiwan. There are approximately 350 species of this genus *Desmodium* worldwide, primarily found in tropical and subtropical regions, with about 48 species recognized for their medicinal properties [8]. In Taiwan’s Flora, eighteen species have been recorded, with sixteen being native [9]. Notably, ten of these species, including *D. caudatum* (Thunb.) DC., *D. gangeticum* (L.) DC., *D. heterocarpon* (L.) DC., *D. heteropyllum* (Willd.) DC., *D. laxiflorum* DC., *D. microphyllum* (Thunb.) DC., *D. multiflorum* DC., *D. renifolium (L.) Schindl.*, *D. sequax* Wall., and *D. triflorum* (L.) DC. are recognized for their medicinal value, often employed in treating various ailments such as cardiovascular syndromes, hepatitis, pneumonia, nephritis, and parasitic diseases. Recent pharmacological studies have revealed the diverse therapeutic effects of *Desmodium* plants, including their antioxidant, anti-inflammatory, cardioprotective, hepatoprotective, sedative, and antipyretic properties [8,10,11,12,13]. Their phytoconstituents include flavonoids, phenylpropanoids, phenolic compounds, and alkaloids, with the former two being the main classes [8,10]. Notably, three commonly used species in Taiwan that were initially classified under the *Desmodium* genus are now listed under different genera in “The Plant List”. The accepted scientific names of these three homotopic synonym species including *D. motorium* (Houtt.) Merr., *D. pulchellum* (L.) Benth., and *D. triquetrum* (L.) DC. are *Codariocalyx motorium* (Houtt.) H. Ohashi., *Phyllodium pulchellum* (L.) Desv., and *Tadehagi triquestrum* (L.) H. Ohashi. in “The Plant List” (http://www.theplantlist.org, accessed on 16 May 2024), respectively. The traditional uses and pharmacological properties of these three species are the same as those of the above-mentioned *Desmodium* species [8,14,15]. Furthermore, Cai’s report has indicated that *D. pulchellum* has MAO inhibitory activity [16]. However, there have been few scientific studies reporting the neuroprotective activities of the *Desmodium* species. Therefore, we collected seven common *Desmodium* plants, including the three homotopic synonym species from Taiwan’s wild low-altitude areas. Initially, this investigation involved comparing the radical scavenging capacities of the seven *Desmodium* species against 2,2-diphenyl-1-picrylhydrazyl (DPPH), 2,2′-azino-bis(3-ethylbenzothiazoline-6-sulphonic acid) (ABTS), and ROS using microtiter spectrophotometric and spectrofluorimetric methods in vitro. As the radical scavenging capacities of plants are closely associated with their reducing power [17], we also compared the relative reducing power of the seven *Desmodium* species. As phenolic compounds, flavonoids, and phenylpropanoids are the major antioxidant phytoconstituents of *Desmodium* species [8,10], the total phenolic, flavonoid, and phenylpropanoid contents in the seven *Desmodium* species were also quantified. Subsequently, the inhibitory activities of the seven *Desmodium* species on MAO-A and MAO-B were further analyzed. We then evaluated and compared the effects of the three *Desmodium* plants with higher antioxidant contents and superior radical scavenging capacities on 6-OHDA auto-oxidation and neuronal damage induced by 6-OHDA and rotenone in SH-SY5Y cells. Finally, we delved into the neuroprotective mechanism of *D. pulchellum* against 6-OHDA-induced neuronal damage in human neuroblastoma SH-SY5Y cells. Our comprehensive study aims to elucidate the neuroprotective potential of the *Desmodium* species, shedding light on their therapeutic applications in the context of neurological disorders, especially PD.

## 2. Results

### 2.1. Antioxidant Phytoconstituent Contents of Seven Desmodium Plants

Phenolic compounds represent a significant class of secondary metabolites found widely across plant species. These compounds endow plants with antioxidant properties and various biological activities, including anti-inflammatory and neuroprotective effects [18]. Consequently, the total phenolic content serves as an important indicator of a medicinal plant’s biological activity and potential medicinal value. Folin–Ciocalteu’s phenol (FCP) method, based on the formation of blue complexes through redox reactions, offers high sensitivity and accuracy for the spectrophotometric determination of phenolic compounds [17,19]. (+)-Catechin, a member of the flavan-3-ol subclass, was used as a reference standard. Over the concentration range of 0–250 μg/mL, a highly linear relationship between concentration and absorbance was observed (y = 0.0097x − 0.0026, R^2^ = 0.998). The absorbance values of all the methanolic filtrates of the *Desmodium* plants (5 mg/mL) fell within this linear range. Upon conversion using the provided linear formula, the total phenolic content present in the *Desmodium* plants is depicted in Figure 1A. Among the seven *Desmodium* plants, *D. pulchellum* contained the highest total phenolic content, followed by *D. motorium* and *D. sequax*, while *D. gangeticum* displayed the lowest content.

Similar to other phenolic compounds, flavonoids are renowned for their antioxidant properties and diverse pharmacological activities [20,21]. The total flavonoid content also serves as a significant parameter in assessing the potential medicinal value of a plant. Aluminum solution reacts with flavonoids such as flavonols or flavan-3-ol to form an aluminum–flavonoid complex through ion chelating reactions [22]. The flavonol compound quercetin was used as a reference standard. There is a highly linear relationship between concentration (0–250 μg/mL) and absorbance which was observed (y = 0.0025x − 0.0057, R^2^ = 0.999). The absorbance values of all the methanolic filtrates of the *Desmodium* plants (10 mg/mL) were within the calibration linear range of quercetin. Based on the provided linear formula, the total flavonoid content present in the *Desmodium* plants is depicted in Figure 1B. *D. motorium* contained the highest total flavonoid content, followed by *D. pulchellum* and *D. triflorum*, while *D. triquetrum* displayed the lowest content.

Phenylpropanoids are a class of secondary metabolites synthesized from phenylalanine and tyrosine through the shikimic acid pathway, serving as intermediate compounds in the biosynthesis of various plant defense secondary metabolites such as flavonoids, coumarins, lignins, and stilbenes. Similarly to phenolic compounds and flavonoids, phenylpropanoids are also recognized for their antioxidant properties and diverse pharmacological activities [23]. Arnow reagent, comprising a mixture of sodium molybdate and sodium nitrite, reacts with phenylpropanoids (dihydroxycinnamic derivatives) [24]. The phenylethanoid glycoside verbascoside was used as a reference standard. There is a highly linear relationship between concentration (0–100 μg/mL) and absorbance which was observed (y = 0.0007x + 0.0245, R^2^ = 0.993). The absorbance values of all the methanolic filtrates of *Desmodium* plants (1 mg/mL) were within the calibration linear range of quercetin. According to the above linear formula, the total phenylpropanoid content present in the *Desmodium* plants is depicted in Figure 1C. *D. pulchellum* contained the highest total phenylpropanoid content, followed by *D. motorium* and *D. sequax*, while *D. gangeticum* displayed the lowest content.

### 2.2. Radical Scavenging Capacities of Seven Desmodium Plants

The DPPH radical is a stable nitrogen radical which has a maximum absorption at 517 nm, resulting in a deep violet color. When an electron or hydrogen atom is donated from plant extracts or natural products, the DPPH solution turns colorless or pale yellow; consequently, the absorbance value at 517 nm decreases. This assay has been widely used for assessing the antioxidant activity of plant extracts or natural products [25]. In this investigation, the absorbance of the DPPH radical (300 μM) at 517 nm was approximately 0.92. The reference standard (+)-Catechin exhibited significant scavenging ability for the DPPH radicals at concentrations ranging from 0 to 25 μg/mL. A strong linear relationship between concentration and scavenging ability was observed (R^2^ = 0.995), with an IC_50_ value of 14.09 ± 0.17 μg/mL for (+)-Catechin against the DPPH radicals. The scavenging percentages of all the methanolic filtrates of *Desmodium* plants (at 0–5 mg/mL) against the DPPH radicals also displayed a concentration-dependent linear relationship (R^2^ = 0.982–0.995). Based on the relative conversion of the *Desmodium* plants and reference standard (+)-catechin at the concentration that provides 50% DPPH scavenging, the DPPH scavenging capacities of these *Desmodium* plants relative to (+)-catechin (i.e., (+)-catechin equivalent of the DPPH radical scavenging capability, CEDSC) are shown in Figure 2A. *D. sequax* exhibited the highest DPPH radical scavenging capacity, followed by *D. pulchellum* and *D. motorium*, while *D. gangeticum* demonstrated the lowest capacity.

The ABTS method commonly employs Trolox as a positive control and compares the reactivity of test antioxidants with it; hence, it is often referred to as the Trolox equivalent antioxidant capacity (TEAC) assay. This assay is also widely used to evaluate the antioxidant activity of crude plant extracts or purified compounds [25,26]. In this investigation, the absorbance of ABTS radical (8 mM) at 734 nm was approximately 0.73. The reference standard Trolox exhibited effective scavenging ability for the ABTS radicals at concentrations ranging from 0 to 50 µM, showing a highly linear relationship between concentration and scavenging ability (R^2^ = 0.999), with an IC_50_ value of Trolox for the ABTS radicals of 28.61 ± 1.24 μM. The scavenging percentages of all the methanolic filtrates of the *Desmodium* plants (at 0–5 mg/mL) for the ABTS radicals also showed a concentration-dependent linear relationship (R^2^ = 0.985–0.993). According to the equivalent activity of the *Desmodium* plants and the reference standard Trolox at concentrations providing 50% ABTS scavenging, the ABTS scavenging capacities of the *Desmodium* plants relative to Trolox (TEAC values) are illustrated in Figure 2B. *D. pulchellum* exhibited the highest ABTS radical scavenging capacity, followed by *D. sequax* and *D. motorium*, while *D. gangeticum* displayed the lowest capacity.

The relative reducing power (RRP) assay relies on the electron-donating activity of plant extracts, natural products, or compounds, thereby facilitating the reduction of Fe^3+^ [17]. In this investigation, the RRP of the reference standard ascorbic acid exhibited a strong linear relationship within the concentration range of 0–20 µM (R^2^ = 0.995). Similarly, the RRP of all the methanolic filtrates of the *Desmodium* plants also displayed a concentration-dependent linear relationship within the concentration range of 0–2.5 mg/mL (R^2^ = 0.990–0.999). Comparing the linear slope between the concentration and absorbance of the *Desmodium* plants with the reference standard (ascorbic acid), the RRP values of these *Desmodium* plants relative to ascorbate were determined, as illustrated in Figure 2C. *D. pulchellum* exhibited the highest reducing power, followed by *D. sequax* and *D. motorium*, while *D. gangeticum* displayed the lowest capacity.

ROS are the byproducts of normal oxygen metabolism in cells and organisms. Due to their high reactivity as free radicals, they can readily induce oxidative damage to cells and tissues. ROS typically encompass superoxide, hydrogen peroxide, and hydroxyl radical. Superoxide, an initial ROS, is primarily generated through intracellular metabolic reactions, notably from the mitochondrial respiratory chain—particularly complex I and complex III—as well as from enzymatic reactions such as the conversion of xanthine to uric acid and hydrogen peroxide by xanthine oxidase in various cell types [27]. In this investigation, superoxide production was simulated by mimicking the physiological reaction between xanthine and xanthine oxidase, quantified by the absorbance value of formazan blue dye resulting from the reduction of nitroblue tetrazolium (NBT) through generated superoxide. As superoxide is primarily detoxified through dismutation by intracellular superoxide dismutase (SOD), SOD serves as a reference standard for assessing superoxide scavenging capacity. The reduction rate of NBT through the reaction of xanthine and xanthine oxidase was approximately 30.5. The reference standard SOD exhibited effective scavenging ability for superoxide at concentrations ranging from 0 to 500 mU/mL, showing a strong logarithmic relationship between concentration and scavenging ability (R^2^ = 0.985), with an IC_50_ value of 144.74 ± 5.66 mU/mL. The methanolic filtrates of the *Desmodium* plants (at 0–10 mg/mL) also displayed concentration-dependent scavenging activities for superoxide, with a logarithmic relationship (R^2^ = 0.981–0.993). According to the equivalent activity of the *Desmodium* plants and reference standard SOD at the concentration that provides 50% superoxide scavenging, the superoxide scavenging capacities of these *Desmodium* plants relative to SOD (SOD equivalent) are illustrated in Figure 3A. *D. sequax* exhibited the highest superoxide scavenging capacity, followed by *D. pulchellum* and *D. motorium*, while *D. gangeticum* showed the lowest capacity.

Hydrogen peroxide, primarily formed from the monovalent reduction of superoxide by SOD or the divalent reduction of oxygen by xanthine oxidase, is relatively inert and non-cytotoxic at or below concentrations of about 20–50 μM. However, excessive hydrogen peroxide entering cells can lead to the generation of the highly reactive hydroxyl radical in the presence of metal ions, potentially causing cellular damage [28]. In this investigation, the hydrogen peroxide scavenging capacities of the methanolic filtrate of *Desmodium* plants were assessed based on the H_2_O_2_-dependent oxidation of homovanillic acid (3-methoxy-4-hydroxyphenylacetic acid, HVA) to a highly fluorescent dimer (2,2′-dihydroxy-3,3′-dimethoxydiphenyl-5,5′-diacetic acid), catalyzed by horseradish peroxidase. The fluorescence intensity of HVA in the presence of hydrogen peroxide (500 µM) at 425 nm was approximately 24,750. The reference standard Trolox exhibited effective scavenging ability for hydrogen peroxide at concentrations ranging from 0 to 100 μM, showing a strong linear relationship between concentration and scavenging ability (R^2^ = 0.997), with an IC_50_ value of Trolox for the hydrogen peroxide radicals of 61.02 ± 1.22 μM. The methanolic filtrates of the *Desmodium* plants (at 0–5 mg/mL) also exhibited concentration-dependent scavenging activities for hydrogen peroxide with a linear relationship (R^2^ = 0.985–0.989). Based on the relative conversion of the *Desmodium* plants and reference standard Trolox at the concentration that provides 50% hydrogen peroxide scavenging, the hydrogen peroxide scavenging capacities of these *Desmodium* plants relative to Trolox (Trolox equivalent) are illustrated in Figure 3B. *D. motorium* exhibited the highest hydrogen peroxide scavenging capacity, followed by *D. triquetrum* and *D. pulchellum*, while *D. gangeticum* showed the lowest capacity.

Finally, the hydroxyl radical—a highly reactive oxygen species in cells—is primarily formed through the Fenton reaction, where hydrogen peroxide reacts with metal ions such as copper or iron. It attacks lipids, polypeptides, proteins, and nucleic acids in the cell membrane, cytosol, and nucleus to cause cell damage and death [29]. Due to the high reaction of hydroxyl radicals, the scavenging activity of the hydroxyl radical is a necessary item in evaluating the ROS scavenging activity of plants and natural products. Therefore, the Fenton reaction was established in vitro with hydrogen peroxide and ferrous sulfate. Luminol chemiluminescence is a very sensitive method for the monitoring of hydroxyl radicals [30]. In this investigation, the luminescence intensity of luminol (2 mM) in the Fenton reaction of hydrogen peroxide and ferrous sulfate was approximately 20,816. The reference standard Trolox exhibited effective scavenging ability for the hydroxyl radicals at concentrations ranging from 0 to 500 mU/mL, showing a strong logarithmic relationship (R^2^ = 0.989), with an IC_50_ value of Trolox for the hydroxyl radicals of 19.79 ± 0.91 μg/mL. The scavenging percentages of all the methanolic filtrates of the *Desmodium* plants (at 0–5 mg/mL) for the hydroxyl radical also presented a Log-linear relationship (R^2^ = 0.971–0.992). According to equivalent activity of the *Desmodium* plants and reference standard Trolox at the concentration that provide 50% hydroxyl radical scavenging, the hydroxyl radical scavenging capacities of these *Desmodium* plants relative to Trolox (Trolox equivalent) are illustrated in Figure 3C. *D. triquetrum* exhibited the highest hydroxyl radical scavenging capacity, followed by *D. pulchellum* and *D. caudatum*, while *D. triflora* showed the lowest capacity.

Additionally, we conducted a Pearson correlation analysis to examine the relationships between the aforementioned antioxidant phytoconstituent contents and radical scavenging capacities across the seven *Desmodium* plants. The results are presented in Table 1, revealing significant positive correlations between certain capacities and contents. Notably, there was a strong positive correlation between total phenolic contents and several radical scavenging capacities, including DPPH radical, ABTS radical, and superoxide (r = 0.835, 0.98, and 0.848; *p* < 0.05, *p* < 0.01, and *p* < 0.05, respectively). These above radical scavenging capacities—encompassing DPPH radical, ABTS radical, and superoxide—also exhibited highly positive correlations with the total phenylpropanoid contents (r = 0.854, 0.903, and 0.83, respectively; all *p* < 0.05). Moreover, the relative reducing power demonstrated strong correlations with total phenolic contents (r = 0.978; *p* < 0.01) and phenylpropanoid contents (r = 0.892; *p* < 0.05). Furthermore, there was a positive and high correlation between the relative reducing power and the aforementioned radical scavenging capacities (DPPH radical, ABTS radical, and superoxide; r = 0.93, 0.995, and 0.939, respectively; all *p* < 0.01).

Subsequently, we calculated the relative reducing ability per milligram of total phenolic contents in the seven *Desmodium* plants, and then analyzed the Pearson correlation and linear relationship between this relative phenolic-reducing ability and radical scavenging capacities. The results, depicted in Figure 4, indicate a significant Pearson correlation and linear relationship between the relative reducing power of total phenolics in the seven *Desmodium* plants and the radical scavenging capacities against the DPPH radical and superoxide (r = 0.965 and 0.945, respectively; both *p* < 0.01).

### 2.3. MAO Inhibitory Activities of Seven Desmodium Plants

The MAO inhibitory activities of the methanolic filtrates of the *Desmodium* plants were assessed using kynuramine as a nonselective substrate. Kynuramine undergoes deamination by MAO to produce 4-hydroxyquinoline, whose fluorescence intensity was measured using a fluorescence reader. To discern the inhibitory effects of the methanolic filtrate of the *Desmodium* plants on MAO-B and MAO-A, the selective MAO-A inhibitor clorgyline and MAO-B inhibitor pargyline were employed. In the present investigation, there was a good linear relationship (R^2^ = 0.999) between the concentration (0–12.5 µM) and fluorescence intensity of 4-hydroxyquinoline. The pretreatment of brain homogenate solely with pargyline (excluding the methanolic filtrate of the *Desmodium* plants) was considered as 100% MAO-A activity. The MAO inhibitory percentages of all the methanolic filtrates of the *Desmodium* plants (at 0–25 mg/mL) for the MAO-A activity exhibited a concentration-dependent relationship (R^2^ = 0.977–0.993). The IC_50_ values for the MAO-A activity of all the methanolic filtrates of the *Desmodium* plants are presented in Table 2. Among the seven *Desmodium* plants, *D. motorium* demonstrated the most potent inhibitory activity on MAO-A, followed by *D. caudatum* and *D. triquetrum*, whereas *D. sequax* exhibited the weakest inhibitory activity on MAO-A. Similarly, the pretreatment of brain homogenate solely with clorgyline (excluding the methanolic filtrate of the *Desmodium* plants) was considered as 100% MAO-B activity. The MAO inhibitory percentages of all the methanolic filtrates of the *Desmodium* plants (at 0–25 mg/mL) for the MAO-B activity also showed a concentration-dependent relationship (R^2^ = 0.985–0.998). The IC_50_ values for the MAO-B activity of all the methanolic filtrates of the *Desmodium* plants are presented in Table 2. *D. sequax* exhibited the most potent inhibitory activity on MAO-B, followed by *D. motorium* and *D. triflorum*, while *D. pulchellum* showed the weakest inhibitory activity on MAO-B. Furthermore, we conducted a ratio analysis of IC_50_ values with respect to MAO-B and MAO-A for the seven *Desmodium* plants in order to confirm their relative selective inhibitory effects on MAO-B and MAO-A. *D. sequax* displayed the best selective inhibitory activity on MAO-B, whereas *D. motorium* exhibited the best selective inhibitory activity on MAO-A.

### 2.4. 6-OHDA Auto-Oxidation Inhibition of Three Desmodium Plants

The compound 6-OHDA, derived from dopamine, is highly unstable and prone to auto-oxidation, resulting in the formation of the intermediate *p*-quinone and ROS such as superoxide and hydrogen peroxide. The toxicity of 6-OHDA is directly linked to the levels of the intermediate *p*-quinone generated during this process. When *p*-quinone is produced from 6-OHDA, it has a maximum absorption at 490 nm. Thus, an analytical method for 6-OHDA auto-oxidation inhibition can serve as an initial assessment to determine whether crude extracts or purified compounds from plants possess protective effects against 6-OHDA-induced neuronal damage [31]. Following the assessment of the radical scavenging capacities of the seven *Desmodium* plants, the three species (*D. pulchellum*, *D. sequax*, and *D. motorium*) exhibiting the highest radical scavenging capacities were selected for further evaluation of their inhibitory effects against 6-OHDA auto-oxidation, with NAC (N-acetylcysteine) serving as a positive control. The inhibitory activities of the three *Desmodium* plants, along with NAC, against 6-OHDA auto-oxidation are depicted in Figure 5. It can be observed that the inhibitory activities of the three *Desmodium* plants against 6-OHDA auto-oxidation were concentration-dependent in the range from 25 to 500 μg/mL. Among them, *D. pulchellum* exhibited the most significant inhibitory effects (*p* < 0.05, *p* < 0.01, *p* < 0.001), followed by *D. motorium* and *D. sequax* (*p* < 0.05, *p* < 0.01, *p* < 0.001). NAC demonstrated remarkable inhibitory effects, achieving inhibition percentages of 83% to 88% at the concentrations of 1 and 2 mM (*p* < 0.001).

### 2.5. Protective Effects against 6-OHDA-Induced or Rotenone-Induced Neurotoxicity in SH-SY5Y Cells

The human neuroblastoma SH-SY5Y cell line, possessing catecholaminergic and neuronal properties which express tyrosine hydroxylase and dopamine β-hydroxylase to synthesize dopamine and noradrenaline, is often regarded as an important in vitro model for investigating the intracellular molecular pathways of neural cells and developing neuroprotective drugs for neurodegenerative diseases, especially Alzheimer’s disease (AD) or PD [32]. Utilizing the dopaminergic neurotoxin 6-OHDA is a common method to establish both in vitro and in vivo PD models, allowing for the assessment of the efficacy of potential treatments derived from plants, natural products, or compounds. Referring to previous studies [33], 25 μM 6-OHDA was used to induce neuronal damage in this study, resulting in a significant reduction in cell viability at 24 h post-treatment (58.4 ± 2.8%; *p* < 0.001) (Figure 6A). Among the tested *Desmodium* plants (*D. pulchellum, D. sequax, and D. motorium*) chosen for their potent radical scavenging capabilities, we observed promising neuroprotective effects against 6-OHDA-induced neuronal damage. NAC was also utilized as a positive control. The neuroprotective activities of the three *Desmodium* plants and NAC against 6-OHDA neuronal damage are shown in Figure 6. Concentration-dependent responses are evident, with all three *Desmodium* species (25–500 μg/mL) exhibiting the ability to mitigate 6-OHDA-induced neuronal damage. Notably, *D. pulchellum* demonstrated the most pronounced reversal of decreased cell viability induced by 6-OHDA (*p* < 0.001), while *D. motorium* and *D. sequax* had the equivalent effects of reversing cell viability (*p* < 0.01, *p* < 0.001). Additionally, NAC showed comparable efficacy in restoring cell viability at the tested concentrations (1 and 2 mM), with its effect at 2 mM equivalent to that of *D. motorium* or *D. sequax* at 500 μg/mL (*p* < 0.001; Figure 6A).

Rotenone, a natural lipophilic pesticide derived from the Fabaceae family, readily penetrates the blood–brain barrier (BBB) to reach the nigrostriatal dopaminergic neuronal pathway. It can specifically induce neurotoxicity in dopaminergic neurons through interrupting the mitochondrial complex I of the electron transport chain, leading to the generation of ROS. We further evaluated the neuroprotective properties of the three *Desmodium* plants against rotenone-induced neuronal damage in the SH-SY5Y cells. Following rotenone exposure for 24 h, cell viability decreased significantly (47.2 ± 1.5%; *p* < 0.001; Figure 6B). Similar to the observations with 6-OHDA, all three *Desmodium* plants (25–500 μg/mL) possessed concentration-dependent neuroprotective effects against rotenone-induced neuronal damage. Once again, *D. pulchellum* demonstrated the most potent reversal of decreased cell viability induced by rotenone (*p* < 0.05, *p* < 0.01, *p* < 0.001), followed by *D. motorium* and *D. sequax* (*p* < 0.01, *p* < 0.001). NAC also demonstrated significant efficacy in restoring cell viability, with its effect at 1 mM equivalent to that of *D. motorium* or *D. sequax* at 500 μg/mL (*p* < 0.001; Figure 6B).

### 2.6. Intracellular ROS Levels and Antioxidant Enzymes Activities in 6-OHDA-Treated SH-SY5Y Cells

As the generation of ROS is an intermediate process in 6-OHDA- or rotenone-induced neurotoxicity, this study employed a non-fluorescent probe, 2′,7′-dichloro-dihydro fluorescein diacetate (DCFH-DA), to label intracellular ROS. DCFH-DA has been widely utilized as a fluorescent probe for detecting intracellular ROS levels as it can freely penetrate cell membranes and is hydrolyzed by intracellular esterases to produce dichloro-dihydro fluorescein (DCFH). Upon the production of intracellular ROS from 6-OHDA or rotenone treatment, DCFH is oxidized by intracellular ROS and converted into the fluorescent compound 2′,7′-fluorescein (DCF). Subsequently, the intensity of DCF fluorescence can be quantified using a fluorescent microplate reader [34]. Given that *D. pulchellum* demonstrated the most significant reversing effects on the decreased cell viability induced by 6-OHDA in the SH-SY5Y cells, we focused solely on evaluating the effects of *D. pulchellum* on the increase in the intracellular ROS levels caused by the 6-OHDA treatment. At 24 h after the 6-OHDA treatment, the fluorescent intensity in the SH-SY5Y cells was enhanced by 6-OHDA to 590.5 ± 37.2%, compared to the control group (100%; see Figure 7). *D. pulchellum* (25–250 μg/mL) effectively attenuated the increase in the intracellular ROS levels caused by 6-OHDA in a concentration-dependent manner, with the highest concentration (250 µg/mL) reducing approximately half of the increase in the fluorescence intensity (*p* < 0.001). NAC at 2 mM resulted in a reduction of approximately 80% in the increase in the fluorescent intensity induced by 6-OHDA (*p* < 0.001; Figure 7).

In general, intracellular ROS levels and antioxidant defense systems maintain a delicate balance to sustain normal cellular function and survival. However, the overproduction of intracellular ROS in response to external infections, internal inflammation, or various stimuli can disrupt this balance, leading to intracellular oxidative stress. ROS indiscriminately attack intracellular molecules such as carbohydrates, nucleic acids, lipids, and proteins, resulting in oxidative damage and alterations in cell structure and function. Ultimately, this oxidative damage can lead to cellular injury and contribute to the development of various diseases, including aging, cancer, and neurodegenerative disorders [35]. Intracellular antioxidant defense systems can be categorized into nonenzymatic and enzymatic systems. Glutathione (GSH), composed of glycine, cysteine, and glutamic acid, is a prominent intracellular nonenzymatic antioxidant. Given its ability to directly scavenge ROS, the intracellular GSH content serves as a crucial indicator of intracellular oxidative stress and the efficacy of antioxidant defense mechanisms [36]. Under the conditions of oxidative stress, lipids, particularly polyunsaturated fatty acids, are highly susceptible to oxidative damage within cells. Malondialdehyde (MDA) serves as a reliable biomarker of cellular oxidative stress, as it is one of the end-products of polyunsaturated fatty acid peroxidation [37]. To assess the impact of antioxidant capacity on the neuroprotective effects of *D. pulchellum* against 6-OHDA-induced neuronal damage, we evaluated the levels of GSH and MDA in the SH-SY5Y cells. Our findings, obtained 24 h after the 6-OHDA treatment, revealed decreased intracellular GSH levels and increased intracellular MDA levels in the 6-OHDA-treated SH-SY5Y cells compared to the control group (*p* < 0.001) (Figure 8). The treatment with *D. pulchellum* (50–250 μg/mL) effectively reversed this phenomenon induced by 6-OHDA in a concentration-dependent manner (*p* < 0.05, *p* < 0.01, *p* < 0.001). Similarly, the treatment with NAC at 2 mM significantly attenuated the changes induced by 6-OHDA (*p* < 0.001; Figure 8).

Superoxide dismutase (SOD), catalase, glutathione peroxidase (GPx), and glutathione reductase (GR) represent the primary enzymatic defense system against intracellular oxidative stress. SOD functions to neutralize superoxide radicals, converting them into hydrogen peroxide (H_2_O_2_), which is further broken down to water by catalase and GPx. To investigate the impact of antioxidant capacity on the neuroprotective effects of *D. pulchellum* against 6-OHDA-induced neuronal damage, we evaluated the activities of SOD, catalase, GPx, and GR in the SH-SY5Y cells. At 24 h following the 6-OHDA treatment, our findings indicated a reduction in antioxidant enzyme activities in the 6-OHDA-treated SH-SY5Y cells compared to the control group (*p* < 0.001; Figure 9). The treatment with *D. pulchellum* (50–250 μg/mL) significantly increased the activities of SOD and GPx in a concentration-dependent manner when compared to the 6-OHDA-treated cells (*p* < 0.05, *p* < 0.01, *p* < 0.001). Notably, a significant elevation in the activities of catalase and GR was observed only with *D. pulchellum* pretreatment at 100–250 μg/mL (*p* < 0.01, *p* < 0.001). Additionally, the treatment with NAC at 2 mM effectively enhanced the antioxidant enzyme activities compared to those observed in 6-OHDA-treated cells (*p* < 0.001; Figure 9).

## 3. Discussion

Based on the previous findings regarding the *Desmodium* genus, various species such as *D. caudatum*, *D. gangeticum*, *D. pulchellum*, and *D. triquestrum* have been associated with cardiovascular protection, liver protection, and anticancer effects due to their antioxidant properties. The primary active phytoconstituents in these species are phenolic compounds, flavonoids, and phenylpropanoids [8,10,14,15]. Our previous report has also highlighted that *D. triflorum* exhibited radical scavenging activities against DPPH and TEAC radicals [38]. Chidambaram et al. have indicated that *D. motorium* exhibited radical scavenging activities against DPPH and TEAC radicals [39]. Another study by Tsai et al. compared the antioxidant phytoconstituents contents and antioxidant activities of 10 *Desmodium* species in Taiwan, identifying that *D. sequax* had the highest total phenolic content and antioxidant activity [40]. To further explore these findings, we collected seven Taiwanese *Desmodium* species, three of which overlapped with Tsai’s study, namely, *D. gangeticum*, *D. sequax*, and *D. triflorum*. Our aim was to compare the antioxidant phytoconstituents, radical scavenging abilities, and relative reducing power among these species. Among the seven common *Desmodium* species, *D. pulchellum* exhibited the highest contents of total phenolics and phenylpropanoids, as well as superior ABTS radical scavenging capacity and relative reducing power. Conversely, *D. motorium* showed the highest flavonoid content but lower total phenolic and phenylpropanoid contents when compared to *D. pulchellum*. Additionally, its radical scavenging capacity against DPPH and ABTS radicals was inferior to that of *D. pulchellum* and *D. sequax*. Although *D. sequax* had lower total phenolic and phenylpropanoid content compared to *D. pulchellum* and *D. motorium*, it displayed the best DPPH radical scavenging capacity. It is worth noting that the total phenolic and flavonoid contents of *D. triflorum* and *D. sequax* differed from those in our previous findings and the values reported by Tsai [38,40]. This variation could be attributed to factors such as harvesting time, collection location, extraction method, and experimental procedures. Furthermore, in our study, the content of phytoconstituents refers to the original medicinal materials of *D. triflorum* and *D. sequax*, whereas in previous studies and Tsai’s report, it pertained to the content of phytoconstituents in the methanolic extract of these species. Considering the oxidative stress hypothesis as a neuropathological mechanism in neurodegenerative disorders, it is crucial to acknowledge that ROS are the primary radicals within cells and are responsible for inducing intracellular oxidative stress. Therefore, we further evaluated the ROS scavenging capacities of the seven common *Desmodium* species. Among the evaluated species, *D. sequax* exhibited the best superoxide scavenging capacity. *D. motorium* had the best hydrogen peroxide scavenging capacity, and *D. triquestrum* had the best hydroxyl radical scavenging capacity. The radical scavenging capacities of *D. pulchellum* on superoxide and hydroxyl radical were inferior to those of *D. sequax* and *D. triquestrum*, respectively.

The above-mentioned techniques used to evaluate the total phytoconstituent contents and radical scavenging capacities of plants or natural products involve the chemical principles of hydrogen atom transfer (HAT) or single electron transfer (SET). For example, the FCP assay and RRP assay predominantly rely on SET mechanisms; conversely, the TEAC assay combines aspects of both SET and HAT mechanisms, and the DPPH assay is primarily based on the SET mechanism, with minor contributions from HAT marginal reactions [25]. Numerous studies have revealed a strong linear correlation between the results of DPPH and ABTS assays and those obtained from the FCP assay. Similarly, a significant correlation has been observed between the RRP and the FCP assays [17,19,25]. The findings of Tsai and Chidambaram have suggested a high correlation between total phenolic contents and ABTS scavenging capacities in *Desmodium* species. Additionally, the ABTS scavenging capacities of *Desmodium* species demonstrated a good linear relationship with the FRAP assay of *Desmodium* species [39,40]. In line with these previous studies, our results also confirmed a high correlation between the total phenolic contents and radical scavenging capacity against DPPH radical, ABTS radical, or superoxide in *Desmodium* species. Moreover, the contents of total phenylpropanoid in the *Desmodium* species also exhibited a noteworthy correlation with the radical scavenging capacity against DPPH radical, ABTS radical, or superoxide in the *Desmodium* species. Furthermore, the radical scavenging capacity against DPPH radical, ABTS radical, or superoxide in *Desmodium* species is closely associated with their reducing ability. Considering that the primary reaction mechanisms of the aforementioned techniques involve SET reactions, the radical scavenging capacity of *Desmodium* species is expected to be linked to their ability in the SET reactions of redox processes. As per Perez’s augmentation regarding the FCP assay [19], phenolic acids, especially phenylpropanoids (hydroxycinnamic acid derivatives), and flavonols exhibit higher reducing properties. In this investigation, we also observed a significant correlation between the total phenolic content of the *Desmodium* species obtained through the FCP method and the phenylpropanoid content of these species obtained using Arnow’s method. Recent research reports on the phytochemical compositions of various *Desmodium* species have indicated that *D. pulchellum* is rich in flavonoids and phenylpropanoids, particularly (-)-catechin, (−)-epicatechin, (−)-gallocatechin, and (−)-epigallocatechin [15]. *D. sequax* has been found to contain phenolic acids (e.g., chlorogenic acid) and flavonoid glycosides (e.g., vitexin) [40]. *D. triflorum* also possesses vitexin derivatives [38]. Additionally, *D. gangeticum* and *D. triquestrum* have been identified as containing flavonoids including the derivatives of quercetin and kaempferol [14,41]. Catechin polyphenols exhibit the highest activity among the above compounds in terms of their radical scavenging abilities against DPPH and ABTS radicals, as indicated in the comparative analysis and previous reports [42,43]. As a result, we propose that *D. pulchellum* exhibits superior radical scavenging capacity and reducing power primarily due to its high contents of catechin polyphenols.

Given the pivotal role of MAO-B as a vital enzyme in dopamine metabolism and oxidative damage in the central dopaminergic system, MAO-B inhibitors stand out as a promising pharmaceutical avenue for treating PD, in contrast to MAO-A inhibitors. Consequently, this investigation also assessed the MAO-A and MAO-B inhibition activities of the *Desmodium* species for comparison purposes. The findings revealed that among the seven *Desmodium* species, *D. motorium* displayed the highest inhibition activity against MAO-A, while *D. sequax* exhibited the most significant inhibition activity against MAO-B. When comparing the IC_50_ of the *Desmodium* species against both MAO-A and MAO-B, *D. sequax* emerged as the most selective inhibitor of MAO-B. Concerning the inhibition activity of *D. pulchellum* against MAO-A and MAO-B, the results were similar to those of Cai [16], indicating a more specific inhibition of MAO-A. Additionally, there was no discernible correlation between the inhibition activity of the *Desmodium* species against MAO-A and MAO-B and their antioxidant components (total phenols and phenylpropanoids). In a prior study, Cai identified alkaloid components such as 5-methoxy-N,N-dimethyltryptamine and N,N-dimethyltryptamine as the primary phytoconstituents responsible for the MAO-A inhibition activity of *D. pulchellum* [16]. In summary, this study marks the initial identification of *D. sequax* from the *Desmodium* species as exhibiting the most selective inhibition of MAO-B, suggesting its potential for further development, while underscoring the need for additional research into its active components.

Based on the findings presented above, we continued our investigation into the neuroprotective properties of two plants (*D. pulchellum* and *D. motorium*) known for their higher antioxidant phytoconstituent contents and one plant (*D. sequax*) recognized for its selective inhibition of MAO-B activity. Adhering to the mitochondrial oxidative stress theory of PD, we conducted experiments utilizing the PD models in the SH-SY5Y neuroblastoma cells through the administration of the dopamine neurotoxin 6-OHDA or the mitochondrial complex-1 inhibitor rotenone, both of which are known to induce neuronal cell damage. Our results demonstrated that all three *Desmodium* plants exhibit concentration-dependent neuroprotective effects against 6-OHDA- or rotenone-induced neuronal damage, with *D. pulchellum* showing the most promising efficacy. Given that the neuronal damage induced by 6-OHDA primarily arises from the intermediate product *p*-quinone and the subsequent ROS generated during auto-oxidation [31], our study revealed that all three *Desmodium* plants also effectively inhibit the 6-OHDA-induced auto-oxidation in a concentration-dependent manner, with *D. pulchellum* exhibiting the most significant impact. Notably, the positive control drug NAC demonstrated neuroprotective effects against 6-OHDA- or rotenone-induced neuronal cell damage, consistent with findings from previous studies [33], and similarly inhibited 6-OHDA auto-oxidation. Furthermore, this study delved into the neuroprotective mechanism of *D. pulchellum* against 6-OHDA-induced neuronal damage based on the mitochondrial oxidative stress neuropathological theory. In line with the existing literature [33], 6-OHDA triggered an increase in the intracellular ROS levels, leading to a reduction in the intracellular non-enzymatic antioxidant GSH levels, the diminished activity of antioxidant enzymes (SOD, catalase, GPx, and GR), and an elevation in the concentration of the lipid oxidation end-product MDA, ultimately culminating in neuronal damage characterized by reduced cell viability. *D. pulchellum*, however, demonstrated the ability to restore the balance in intracellular ROS and antioxidant defense systems disrupted by the 6-OHDA treatment, thereby mitigating the occurrence of neuronal damage. Numerous previous reports have indicated that catechin polyphenols possess protective effects against neuronal damage and apoptosis provoked by 6-OHDA, rotenone, or MPTP [44,45]. This protective mechanism is primarily attributed to their robust ability to scavenge radicals and trigger intracellular antioxidant enzymes, thereby engaging various intracellular signaling pathways. Consolidating our current discoveries with those of fellow researchers [15,16], *D. pulchellum* emerges as a standout species among the seven commonly utilized Taiwanese *Desmodium* species, showcasing notable proficiency in scavenging radicals and exerting neuroprotective effects. Its principal constituents are presumed to be catechin polyphenols, including (-)-catechin, (−)-epicatechin, (−)-gallocatechin, and (−)-epigallocatechin [15]. Nevertheless, it presented slight selectivity towards MAO-A, with its active phytoconstituents possibly encompassing alkaloids such as 5-methoxy-N,N-dimethyltryptamine and N,N-dimethyltryptamine [16]. Our forthcoming endeavors for research on *D. pulchellum* will concentrate on devising a standardized preparation method for its neuroprotective catechin polyphenols, quantifying its primary active ingredients, assessing its neuroprotective efficacy, and delving deeper into its molecular mechanisms. *D. sequax* also exhibited notable radical scavenging ability and neuroprotective effects; moreover, it displayed a significant and selective ability to inhibit MAO-B. These effects are similar to the action of known MAO-B inhibitors such as selegiline and rasagiline [1,3]. Tsai’s report has suggested that *D. sequax* contains chlorogenic acid and vitexin [40]. Chlorogenic acid—but not vitexin—exhibits inhibitory effects on MAO-B [44,46], indicating that the active ingredients might comprise compounds more akin to chlorogenic acid. However, further phytochemical investigation is needed to isolate and confirm the activities of these ingredients.

## 4. Materials and Methods

### 4.1. Plant Collection and Filtrate Preparation

Seven *Desmodium* species including *D. caudatum, D. gangeticum, D. motorium, D. pulchellum, D. sequax, D. triflorum,* and *D. triquetrum* were collected from low-altitude regions in Taiwan and identified by Prof. Hung-Chi Chang from Chaoyang University of Technology. Entire plants of these seven *Desmodium* species were dried and ground into a coarse powder. Subsequently, each powder (1 g) was extracted with 10 mL of methanol and sonicated for 90 min, then filtered using a 0.22 μm filter, and adjusted to a total volume of 10 mL to obtain a methanolic standardized filtrate with a concentration of 100 mg/mL. This filtrate was further diluted with distilled water to evaluate the contents of antioxidant phytoconstituents and radical scavenging activities. For the 6-OHDA auto-oxidation and neurotoxicity experiments, the methanolic standardized filtrate of the *Desmodium* plants was dried with nitrogen, then dissolved again with phosphate-buffered solution (PBS) or Dulbecco’s modified Eagle’s medium (DMEM).

### 4.2. Antioxidant Phytoconstituent Contents

#### 4.2.1. Total Phenolic Contents

The determination of total phenolic contents in the *Desmodium* plants was conducted using a microtiter spectrophotometric reader (PowerWave X340, Bio-Tek Inc., Winooski, VT, USA) following the procedure outlined in our previous report [24]. The methanolic standardized filtrate (5 mg/mL) or a reference standard (+)-catechin (0–250 µg/mL) was combined with 50% (*v*/*v*) FCP reagent and 10% sodium carbonate solution in triplicate. This reaction mixture was shaken for 20 s, then incubated at room temperature (RT) for 30 m. Absorbance was measured at 725 nm. The total phenolic contents in the *Desmodium* plants are expressed as the relative amounts of (+)-catechin per gram of the *Desmodium* plants (mg catechin/g sample).

#### 4.2.2. Total Flavonoid Contents

The total flavonoid contents in the *Desmodium* plants were determined according to our previously reported method [24]. The methanolic standardized filtrate (10 mg/mL) or a reference standard quercetin (0–100 µg/mL) was mixed with 2% (*w*/*v*) aluminum nitrate and 0.2 M potassium acetate in triplicate. This reaction solution was shaken for 20 s, then incubated at RT for 15 min. Absorbance was recorded at 415 nm. The total flavonoid contents in the *Desmodium* plants are expressed as the relative amounts of quercetin per gram of the *Desmodium* plants (mg quercetin/g sample).

#### 4.2.3. Total Phenylpropanoid Contents

The assessed method of total phenylpropanoid contents in the *Desmodium* plants followed our previously reported method [24]. The methanolic standardized filtrate (1 mg/mL) or a reference standard verbascoside (0–100 µg/mL) was mixed with 10% (*w*/*v*) Arnow reagent and 2N sodium hydroxide in triplicate. This reaction solution was shaken for 30 s, then incubated at RT for 10 min. Absorbance was recorded at 525 nm. The total phenylpropanoid contents in the *Desmodium* plants are expressed as the relative amounts of verbascoside per gram of the *Desmodium* plants (mg verbascoside/g sample).

### 4.3. Radical Scavenging Capacities

#### 4.3.1. DPPH Radical Scavenging Capacities

According to our previously reported method [24], the DPPH radical scavenging capacities of the *Desmodium* plants were assessed. The methanolic standardized filtrate (0–5 mg/mL) or a reference standard (+)-catechin (0–25 µg/mL) was pipetted into each well of a 96-well plate and mixed with 300 µM DPPH methanol solution or methanol in triplicate. This reaction solution was shaken for 15 s, then incubated at RT for 30 min in the dark. Absorbance was recorded at 517 nm. The DPPH radical scavenging capacity of the *Desmodium* plants is expressed as reference standard (+)-catechin equivalents in milligrams per gram of the *Desmodium* plants (abbreviated as CEDSC, mg catechin/g sample).

#### 4.3.2. ABTS Radical Scavenging Capacities

The ABTS radical scavenging capacities of the *Desmodium* plants were also assessed following the procedure outlined in our previous report [24]. The methanolic standardized filtrate (0–5 mg/mL) or a reference standard Trolox (0–50 µM) was dispensed into each well of a 96-well plate and mixed with a diluted ABTS solution (mix 8 mM ABTS solution and 8.4 mM potassium persulfate solution in a ratio of 2:1, dilute it 50 times with ethanol the next day) or ethanol in triplicate. This reaction solution was shaken for 15 s, then incubated at RT for 4 min in the dark. Absorbance was recorded at 734 nm. The ABTS radical scavenging capacity of the *Desmodium* plants is expressed as reference standard Trolox equivalents in millimole per gram of the *Desmodium* plants (abbreviated as TEAC, mmol Trolox/g sample).

#### 4.3.3. RRP Assay

The RRP assay of the *Desmodium* plants followed the procedure outlined in Vasyliev’s report [47]. The methanolic standardized filtrate (0–2.5 mg/mL) or a reference standard ascorbic acid (0–20 µM) was dispensed into each well of a 96-well plate and mixed with 1% potassium ferricyanide (dissolved in 50 mM HCl), 20 mM ferric chloride, and 50 mM sodium phosphate-buffered solution (pH 6.6) in triplicate. This reaction solution was then incubated at 50 °C for 20 min and subsequently treated with 1% trichloroacetic acid and 5 mM ferric chloride. Absorbance was recorded at 700 nm. The RRP value of the *Desmodium* plants is expressed as reference standard ascorbic acid equivalents in micromole per gram of the *Desmodium* plants (relative reducing power, abbreviated as RRP, µmol ascorbate/g sample).

#### 4.3.4. Superoxide Scavenging Capacities

The superoxide scavenging capacities of the *Desmodium* plants were evaluated following the methodology outlined in our previous study [31]. In brief, the methanolic standardized filtrate (0–10 mg/mL) or a reference standard SOD (0–500 mU/mL) was added into each well of a 96-well plate, then mixed with a solution containing 800 µM xanthine, 240 mU/mL xanthine oxidase, and 168 µM nitroblue tetrazolium (dissolved in 50 mM PBS; pH 7.4) in triplicate. After 15 s of agitation, the kinetic slope of the absorbance values was measured over a period of 5 min at 560 nm at RT. The superoxide scavenging capacity of the *Desmodium* plants is expressed as reference standard SOD equivalents in units per gram of the *Desmodium* plants (SOD equivalent, U SOD/g sample).

#### 4.3.5. Hydrogen Peroxide Scavenging Capacities

The hydrogen peroxide scavenging capacities of the *Desmodium* plants were also conducted following the procedure outlined in our previous study [31]. Briefly, the methanolic standardized filtrate (0–5 mg/mL) or a reference standard Trolox (0–100 µM) was dispensed into each well of a 96-well plate, then mixed with 500 µM H_2_O_2_, 8 U/mL horseradish peroxidase, and 5 mM homovanillic acid (dissolved in 50 mM PBS (pH 7.4)) in triplicate. This reaction solution was shaken for 15 s, then incubated at RT for 25 min in the dark. Fluorescence intensity was measured using a fluorescence spectrometric reader (FLX800, Bio-Tek Inc., Winooski, VT, USA) at an excitation wavelength of 315 nm and an emission wavelength of 425 nm. The hydrogen peroxide scavenging capacity of the *Desmodium* plants is expressed as reference standard Trolox equivalents in millimole per gram of the *Desmodium* plants (Trolox equivalent, mmol Trolox/g sample).

#### 4.3.6. Hydroxyl Radical Scavenging Capacities

The hydroxyl radical scavenging capacities of the *Desmodium* plants were conducted with modifications based on the procedure outlined in Cheng’s report [48]. The methanolic standardized filtrate (0–5 mg/mL) or a reference standard Trolox (0–500 µM) was dispensed into each well of a 96-well plate, then mixed with 20 mM H_2_O_2_, 3 mM ferrous sulfate (dissolved in 1 mM EDTA solution), and 2 mM luminol salt solution in triplicate. After 15 s of agitation, the reaction mixture was incubated at RT for 1 min in the dark. The intensity of chemiluminescence was recorded. The hydroxyl radical scavenging capacity of the *Desmodium* plants is expressed as reference standard Trolox equivalents in millimole per gram of the *Desmodium* plants (Trolox equivalent, mmol Trolox/g sample).

### 4.4. MAO Inhibitory Activities

The MAO inhibitory activities of the *Desmodium* plants were assessed based on the procedure outlined in Haraguchi’s report [49] with modifications. Initially, brain mitochondrial homogenates were prepared by homogenizing rat cerebral hemispheres in ice-cold 50 mM PBS solution (containing 250 mM sucrose and 0.5 mM EDTA, pH 7.4). This homogenization process was conducted using a Digital Homogenizer, followed by centrifugation with a sucrose concentration gradient in two different solutions (30 and 60 mM sucrose containing 3–6% Ficoll, 120–240 mM mannitol, and 25–50 µM EDTA sequentially, pH 7.4) at 14,000 rpm for 30 min. The resulting mitochondrial homogenate was then suspended in 50 mM PBS solution (pH 7.4). Subsequently, these mitochondrial homogenate suspensions were preincubated with either clorgyline (2 µM) or pargyline (2 µM) to assess MAO-A or MAO-B activity, respectively. Following preincubation, the enzyme solution was transferred into each well of a 96-well plate and mixed with the methanolic standardized filtrate (0–25 mg/mL) and 2.5 mM kynuramine solution in triplicate. This reaction solution was shaken for 15 s, then incubated at 37 °C for 20 min. The termination of the reaction was achieved by adding 10% ZnSO_4_ and 2 N NaOH. The fluorescence of the final product 4-hydroxyquinoline was measured at an excitation wavelength of 315 nm and an emission wavelength of 380 nm. For reference, kynuramine was replaced with 4-hydroxyquinoline as a standard, and a calibration curve of 4-hydroxyquinoline (0–12.5 µM) was established to quantify the amount produced in the reaction described above. The inhibitory percentage of 4-hydroxyquinoline formation caused by the *Desmodium* plants was calculated to determine the *IC*_50_ values of the *Desmodium* plants against MAO-A or MAO-B.

### 4.5. 6-OHDA Auto-Oxidation Inhibition

The 6-OHDA auto-oxidation inhibitory assay of the *Desmodium* plants was conducted according to the procedure outlined in our previous report [31]. This assay was performed in a cell-free system designed to mimic the conditions corresponding to 6-OHDA treatment in cell experiments. In brief, the PBS-redissolved filtrate (0–500 µg/mL) or a reference standard NAC (1 or 2 mM) was pipetted into each well of a 96-well plate and mixed with 25 µM 6-OHDA (dissolved in 10 mM PBS solution) in triplicate. This reaction solution was shaken for 15 s at 37 °C in the dark. Subsequently, the absorbance at 490 nm, which reflects the levels of the formed intermediate product *p*-quinone was monitored for 3 min at 30 s intervals at 37 °C.

### 4.6. Protection against 6-OHDA-Induced Neurotoxicity in SH-SY5Y Cells

#### 4.6.1. Cell Culture

Human neuroblastoma SH-SY5Y cells were procured from the American Type Culture Collection (ATCC; Manassas, VA, USA). Following the guidelines provided by the ATCC, the cells were thawed and cultured in a 25 cm^3^ culture flask using the DMEM medium, supplemented with 10% FBS, penicillin (100 U/mL), and streptomycin (100 μg/mL). The cells were incubated in a water-saturated atmosphere incubator with 5% CO_2_, and the culture medium was replenished every 2–3 days once they reached 70−80% confluency. Upon achieving confluency, the cells were washed with a fresh 0.25% trypsin solution (containing 0.53 mM EDTA), then seeded into 96-well sterile clear-bottom plates (2 × 10^4^ cells/well), 6-well sterile clear-bottom plates (8 × 10^5^ cells/well), or 90 mm sterile clear-bottom dishes (4 × 10^6^ cells/dish). The cell experiments were conducted 24 h after seeding.

#### 4.6.2. Estimation of Cell Viability

Following exposure to 6-OHDA (25 μM) or rotenone (1 μM) for 24 h [33,50], the viability of the cells was evaluated using the MTT assay in accordance with our previous protocols [31]. To assess the protective effects of the methanolic extracts from three *Desmodium* plant species, namely *D. pulchellum*, *D. sequax*, and *D. motorium*, on the SH-SY5Y cells, a 15 min treatment was administered prior to the addition of 6-OHDA or rotenone. NAC was utilized as a positive control, which was also pretreated before the addition of 6-OHDA or rotenone [33]. The methanolic standardized filtrate from the three *Desmodium* plants was dried with nitrogen and subsequently reconstituted with DMEM. The solution was then filtered using a 0.22 μM sterile filter to obtain the stock solution of the methanolic extracts. The working solution (25–500 μg/mL) was freshly prepared for experimentation.

#### 4.6.3. Estimation of Intracellular ROS Levels

The intracellular ROS levels were assessed using the ROS-sensitive cell-permeant fluorophore DCFH-DA (Sigma-Aldrich Chemical Co., St. Louis, MO, USA) following our established protocol [31]. The SH-SY5Y cells were seeded into clear-bottom black 96-well culture plates and subjected to treatment with 6-OHDA (with or without pretreatment with the methanolic extract of *Desmodium pulchellum* or NAC) for 24 h. Subsequently, DCFH-DA (100 μM) was added to the cells and incubated for 30 min at 37 °C in the dark. The cells were then washed with PBS and placed in DMEM without phenol red. The fluorescence intensity was measured at Ex 485/Em 530 nm using a fluorescent microplate reader. The data are expressed as a percentage relative to untreated cells, which served as the control group (designated as 100%).

#### 4.6.4. Estimation of Intracellular Antioxidant Makers

The activities of intracellular antioxidant makers, such as SOD and catalase, as well as the levels of GSH and MDA, were assessed following the protocol described in our previous publication [51]. The activities of GPx and GR were determined using commercially available enzyme-linked immunosorbent assay (ELISA) kits (Cayman Chemical Company, Ann Arbor, MI, USA), according to the manufacturer’s instructions. The SH-SY5Y cells were seeded into 90 mm sterile clear-bottom dishes and treated with 6-OHDA (with or without pretreatment with the methanolic extract of *Desmodium pulchellum* or NAC) for 24 h. Subsequently, the cells were collected, sonicated, and centrifuged for 15 min at 4 °C. The resulting cell supernatant was utilized to assay the aforementioned antioxidant markers. The activities of SOD are expressed as U per milligram of protein, while the activities of GR are expressed as mU per milligram of protein. The activities of GPx are expressed as μmol NADPH oxidation/min/milligram of protein. The catalase activities are expressed as μmol H_2_O_2_ degradation/min/milligram of protein. The levels of GSH and MDA were expressed as nmol per milligram of protein.

### 4.7. Statistical Analysis

The data of the phytoconstituent contents, radical scavenging capacities, MAO inhibitory activities, and 6-OHDA auto-oxidation inhibitory activities obtained from three repeated experiments are expressed as mean ± standard deviation (SD). Similarly, the data of the cell viability, intracellular ROS levels, and oxidative markers, obtained from four repeated experiments, are also presented as mean ± SD. Statistical analysis was performed using one-way analysis of variance (ANOVA), followed by Turkey’s test, using the statistical software SPSS 20.0 for Windows. Probability values less than 0.05 were considered statistically significant.

## 5. Conclusions

This study compared the radical scavenging activity, reducing power, and MAO inhibitory activity of seven Taiwanese *Desmodium* species. Among them, *D. pulchellum*, *D. sequax*, and *D. motorium* showed the highest radical scavenging activity and reducing power, effectively neutralizing superoxide anions. Notably, *D. sequax* demonstrated the most selective inhibitory effect on MAO-B activity. Their radical scavenging activity was primarily correlated with their total phenolic contents, particularly the phenylpropanoids, while the MAO inhibitory activity was not reliant on these compounds. Additionally, this study unveiled that the aforementioned three *Desmodium* species, especially *D. pulchellum*, exhibit neuroprotective effects against 6-OHDA-induced neurotoxicity. *D. pulchellum* was observed to counteract oxidative damage and restore the intracellular antioxidant defense system impaired by 6-OHDA. Consequently, we suggest that *D. pulchellum*, among the assessed *Desmodium* plants, has the potential to be developed into a therapeutic drug for Parkinson’s disease, which is likely due to its abundance of total phenolic compounds.

## Figures and Tables

**Figure 1 plants-13-01742-f001:**
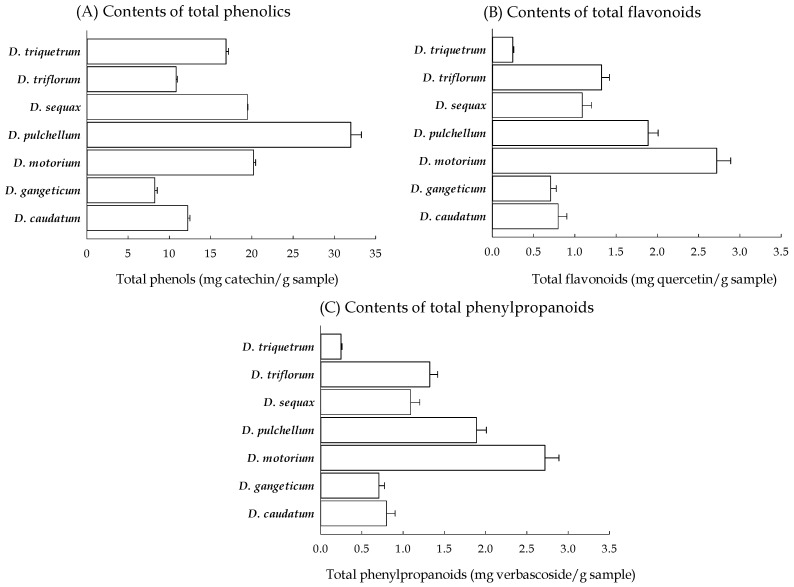
The antioxidant phytoconstituent contents of the methanolic filtrates of the seven *Desmodium* plants. (**A**) Total phenolic contents; (**B**) total flavonoid contents; (**C**) total phenylpropanoid contents. *D. caudatum*: *Desmodium caudatum* (Thunb.) DC.; *D. gangeticum*: *Desmodium gangeticum* (L.) DC.; *D. motorium*: *Desmodium motorium* (Houtt.) Merr.; *D. pulchellum*: *Desmodium pulchellum* (L.) Benth.; *D. sequax*: *Desmodium sequax* Wall.; *D. triflorum*: *Desmodium triflorum* (L.) DC.; *D. triquetrum*: *Desmodium triquetrum* (L.) DC.

**Figure 2 plants-13-01742-f002:**
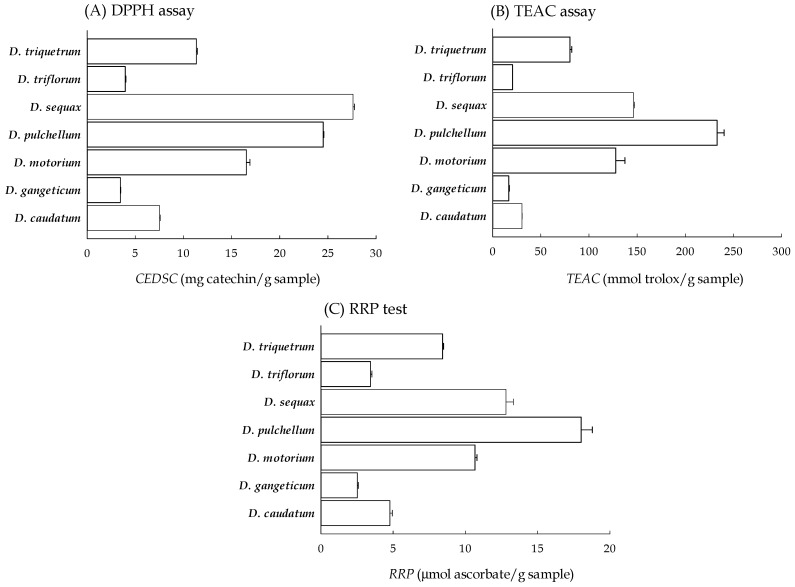
The radical scavenging capacities of the methanolic filtrates of the seven *Desmodium* plants. (**A**) DPPH assay; (**B**) TEAC assay; (**C**) RRP test. *D. caudatum*: *Desmodium caudatum* (Thunb.) DC.; *D. gangeticum*: *Desmodium gangeticum* (L.) DC.; *D. motorium*: *Desmodium motorium* (Houtt.) Merr.; *D. pulchellum*: *Desmodium pulchellum* (L.) Benth.; *D. sequax*: *Desmodium sequax* Wall.; *D. triflorum*: *Desmodium triflorum* (L.) DC.; *D. triquetrum*: *Desmodium triquetrum* (L.) DC.; CEDSC: (+)-catechin equivalent of DPPH radical scavenging capacity; DPPH: 1,1-diphenyl-2-picryhydrazyl; RRP: relative reducing power; TEAC: Trolox equivalent antioxidant capacity.

**Figure 3 plants-13-01742-f003:**
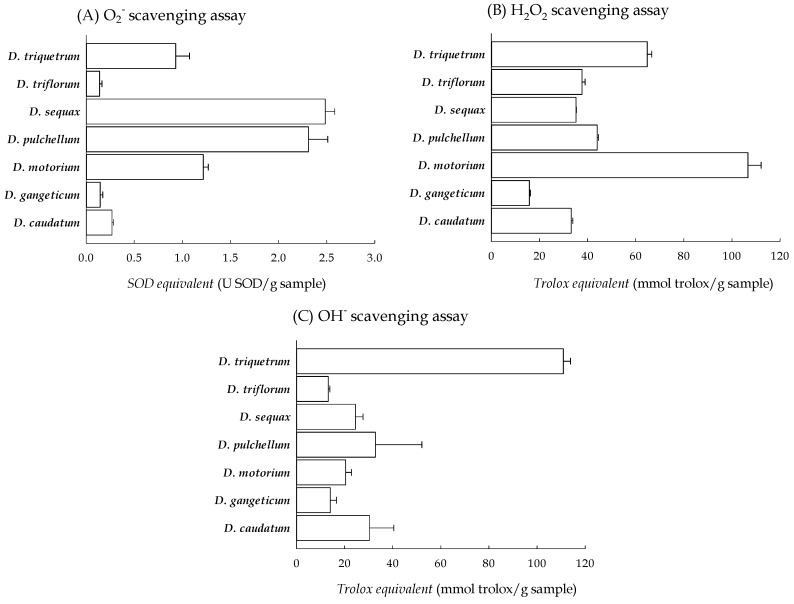
The ROS scavenging capacities of the methanolic filtrates of the seven *Desmodium* plants. (**A**) O_2_^−^ scavenging assay; (**B**) H_2_O_2_ scavenging assay; (**C**) OH^−^ scavenging assay. *D. caudatum*: *Desmodium caudatum* (Thunb.) DC.; *D. gangeticum*: *Desmodium gangeticum* (L.) DC.; *D. motorium*: *Desmodium motorium* (Houtt.) Merr.; *D. pulchellum*: *Desmodium pulchellum* (L.) Benth.; *D. sequax*: *Desmodium sequax* Wall.; *D. triflorum*: *Desmodium triflorum* (L.) DC.; *D. triquetrum*: *Desmodium triquetrum* (L.) DC.; SOD: superoxide dismutase.

**Figure 4 plants-13-01742-f004:**
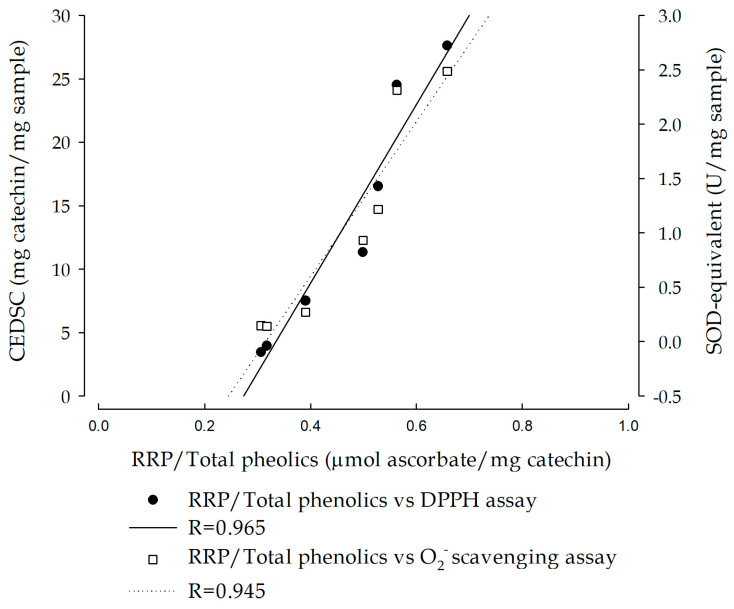
The linear relationship between the relative reducing power of total phenolics and the radical scavenging capacities of the methanolic filtrates of the seven *Desmodium* plants. CEDSC: (+)-catechin equivalent of DPPH radical scavenging capacity; DPPH: 1,1-diphenyl-2-picryhydrazyl; RRP: relative reducing power.

**Figure 5 plants-13-01742-f005:**
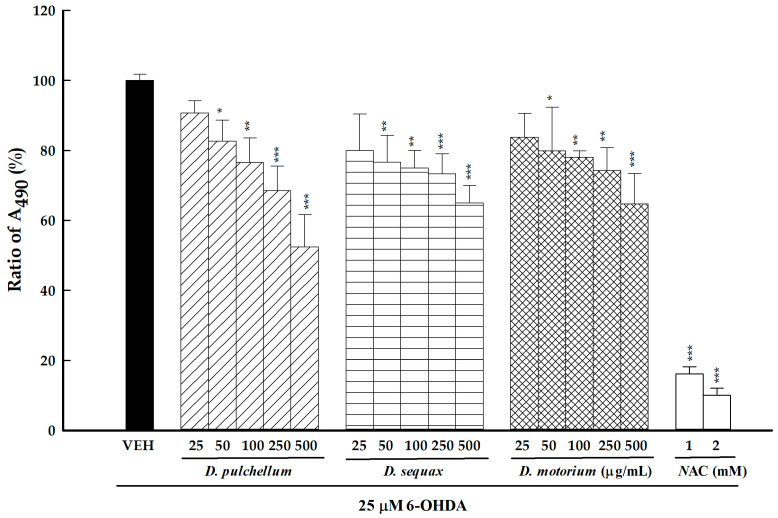
Inhibitory activities of the methanolic extracts of the three *Desmodium* plants (25, 50, 100, 250, and 500 μg/mL) and NAC (1 and 2 mM) against 6-OHDA (25 μM) auto-oxidation. The results are expressed as the mean ± SD (n = 3). * *p* < 0.05, ** *p* < 0.01, and *** *p* < 0.001, compared with the VEH/6-OHDA group. *D. motorium*: *Desmodium motorium* (Houtt.) Merr.; *D. pulchellum*: *Desmodium pulchellum* (L.) Benth.; *D. sequax*: *Desmodium sequax* Wall.; 6-OHDA: 6-hydroxydopamine; NAC: N-acetylcysteine; VEH: vehicle.

**Figure 6 plants-13-01742-f006:**
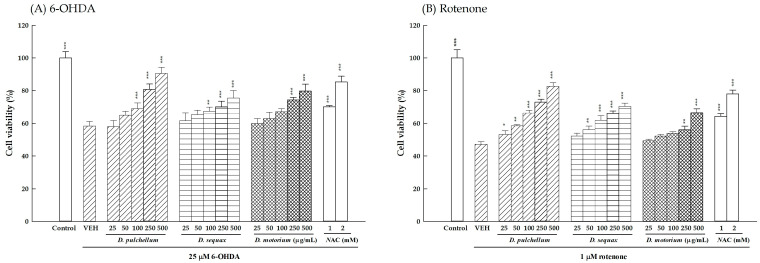
The protective effects of the methanolic extracts of the three *Desmodium* plants (25, 50, 100, 250, and 500 μg/mL) and NAC (1 and 2 mM) against 6-OHDA (25 μM)-induced or rotenone (1 μM)-induced neurotoxicity in the SH-SY5Y cells. (**A**) 6-OHDA; (**B**) rotenone. The results are expressed as the mean ± SD (n = 4). * *p* < 0.05, ** *p* < 0.01, and *** *p* < 0.001, compared with the VEH/6-OHDA group. *D. motorium*: *Desmodium motorium* (Houtt.) Merr.; *D. pulchellum*: *Desmodium pulchellum* (L.) Benth.; *D. sequax*: *Desmodium sequax* Wall.; 6-OHDA: 6-hydroxydopamine; NAC: N-acetylcysteine; VEH: vehicle.

**Figure 7 plants-13-01742-f007:**
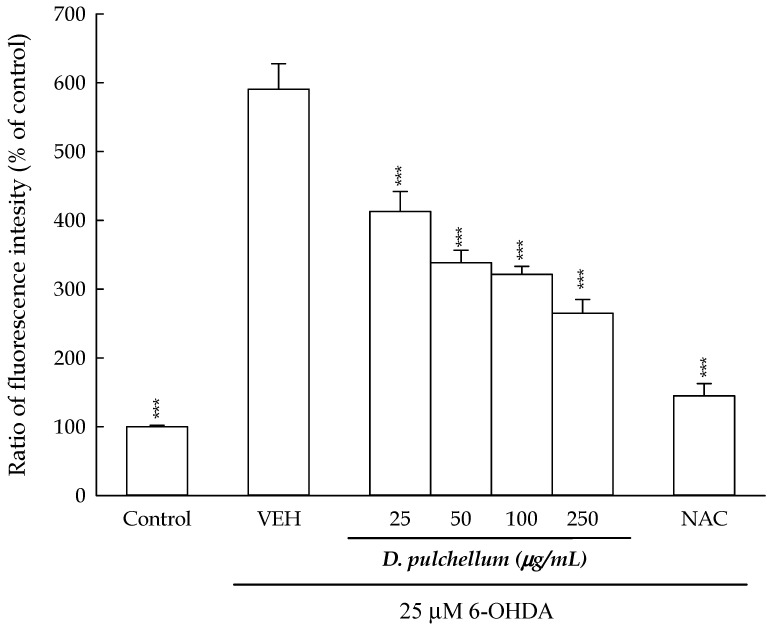
Effects of the methanolic extracts of *Desmodium pulchellum* (25, 50, 100, and 250 μg/mL) and NAC (2 mM) on the intracellular ROS levels in the 6-OHDA (25 μM)-treated SH-SY5Y cells. The results are expressed as the mean ± SD (n = 4). *** *p* < 0.001, compared with the VEH/6-OHDA group. *D. pulchellum*: *Desmodium pulchellum* (L.) Benth.; 6-OHDA: 6-hydroxydopamine; NAC: N-acetylcysteine; VEH: vehicle.

**Figure 8 plants-13-01742-f008:**
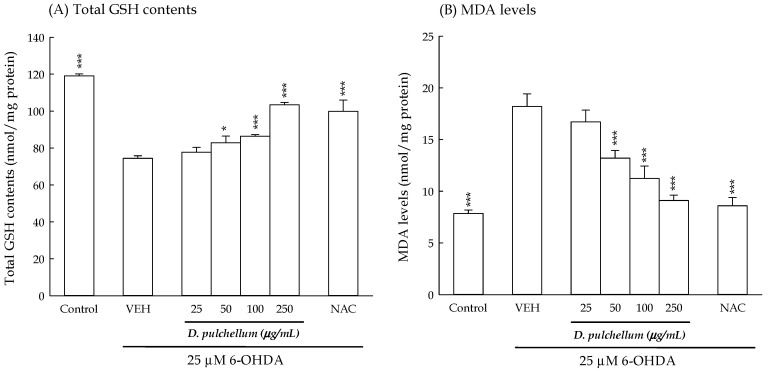
Effects of the methanolic extracts of *Desmodium pulchellum* (25, 50, 100, and 250 μg/mL) and NAC (2 mM) on the intracellular GSH contents and MDA levels in the 6-OHDA (25 μM)-treated SH-SY5Y cells. (**A**) GSH contents; (**B**) MDA levels. The results are expressed as the mean ± SD (n = 4). * *p* < 0.05, *** *p* < 0.001, compared with the VEH/6-OHDA group. *D. pulchellum*: *Desmodium pulchellum* (L.) Benth.; 6-OHDA: 6-hydroxydopamine; GSH: glutathione; MDA: malondialdehyde; NAC: N-acetylcysteine; VEH: vehicle.

**Figure 9 plants-13-01742-f009:**
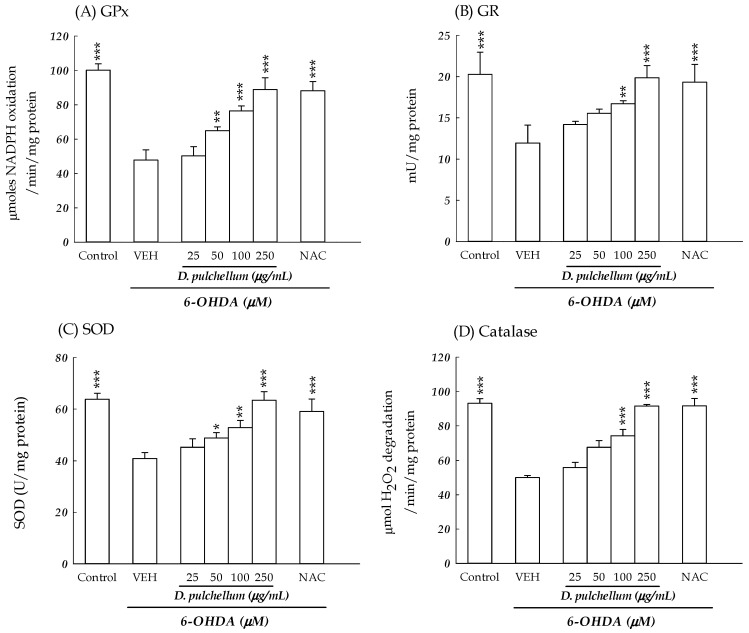
Effects of the methanolic extracts of *Desmodium pulchellum* (25, 50, 100, and 250 μg/mL) and NAC (2 mM) on the activities of intracellular antioxidant enzymes in the 6-OHDA (25 μM)-treated SH-SY5Y cells. (**A**) GPx; (**B**) GR; (**C**) SOD; (**D**) catalase. The results are expressed as the mean ± SD (n = 4). * *p* < 0.05, ** *p* < 0.01, and *** *p* < 0.001, compared with the VEH/6-OHDA group. *D. pulchellum*: *Desmodium pulchellum* (L.) Benth.; 6-OHDA: 6-hydroxydopamine; GPx: glutathione peroxidase; GR: glutathione reductase; NAC: N-acetylcysteine; SOD: superoxide dismutase; VEH: vehicle.

**Table 1 plants-13-01742-t001:** Pearson correlation coefficients (r) between the contents of antioxidant phytoconstituents and radical scavenging capacities of the *Desmodium* plants.

	DPPH	TEAC	O_2_^−^	H_2_O_2_	OH^−^	RRP	TP	TF
TEAC	0.912 **							
O_2_^−^	0.992 **	0.925 **						
H_2_O_2_	0.249	0.33	0.195					
OH^−^	0.033	0.068	0.064	0.252				
RRP	0.93 **	0.995 **	0.939 **	0.334	0.13			
TP	0.835 *	0.98 **	0.848 *	0.362	0.139	0.978 **		
TF	0.393	0.525	0.349	0.628	−0.502	0.475	0.52	
TPPs	0.854 *	0.903 *	0.83 *	0.537	−0.169	0.892 *	0.876 **	0.793 *

* *p* < 0.05, ** *p* < 0.01. DPPH: 1,1-diphenyl-2-picryhydrazyl; RRP: relative reducing power; TEAC: Trolox equivalent antioxidant capacity; TF: total flavonoids; TP: total phenolics; TPPs: total phenylpropanoids.

**Table 2 plants-13-01742-t002:** The MAO inhibitory activities of the methanolic filtrates of the seven *Desmodium* plants.

Samples	IC_50_ in MAO-A (mg/mL)	IC_50_ in MAO-B (mg/mL)	Ratio (MAO-B/MAO-A)
*D. caudatum*	2.44 ± 0.25	4.41 ± 0.14	1.81
*D. gangeticum*	3.75 ± 0.10	5.45 ± 0.13	1.45
*D. motorium*	1.10 ± 0.10	2.31 ± 0.04	2.11
*D. pulchellum*	3.00 ± 0.08	6.23 ± 0.15	2.08
*D. sequax*	7.22 ± 0.67	1.13 ± 0.01	0.16
*D. triflorum*	4.29 ± 0.28	4.19 ± 0.11	0.98
*D. triquetrum*	2.94 ± 0.10	5.21 ± 0.08	1.77

Data were expressed as mean ± SD (n = 3). *D. caudatum*: *Desmodium caudatum* (Thunb.) DC.; *D. gangeticum*: *Desmodium gangeticum* (L.) DC.; *D. motorium*: *Desmodium motorium* (Houtt.) Merr.; *D. pulchellum*: *Desmodium pulchellum* (L.) Benth.; *D. sequax*: *Desmodium sequax* Wall.; *D. triflorum*: *Desmodium triflorum* (L.) DC.; *D. triquetrum*: *Desmodium triquetrum* (L.) DC.

## Data Availability

Data are contained within the article.

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
