# Peer review of "Exploring the Neuroprotective Potential of *Desmodium* Species: Insights into Radical Scavenging Capacity and Mechanisms against 6-OHDA-Induced Neurotoxicity"

_plants, 2024, doi:10.3390/plants13131742_

Round 1

Reviewer 1 Report

Comments and Suggestions for Authors

This manuscript presents an extensive study of the neuroprotective potential of extracts obtained from seven plants of the genus Desmonium from Taiwan.

The chemical characterization includes the determination of phenols, flavonoids and total phenylpropanoids in the methanol extracts of these plants. The biological activities evaluated include antioxidant activity, free radical scavenging capacity, inhibition of monoamine oxidase and 6-hydroxydopamine autooxidation. Based on the results, for the most promising species, the study of neuroprotective activity in SH-SY5Y cells was undertaken and the study of neuroprotection mechanisms was deepened.

The results of the study lead the authors to propose that the species Desmonium pulchellum has the potential to develop a drug for the treatment of Parkinson's disease.

Although some of the results presented in this manuscript partially overlap with other already published works, as a whole the study is broader and more complete, and delves into the neuroprotective activity of the chosen plants in general, and of D. pulchellum in particular. 

I wonder if, for the sake of brevity, the authors could eliminate or summarize some of the descriptions included in the Results section for the determinations made. In some cases they are well-known methods in Natural Products studies (for example total phenols or total flavonoids).

Finally, in my opinion the manuscript has its merits and is worth publishing.

Comments on the Quality of English Language

A few typos are detected in the manuscript, for example on page 3, line 132 and on page 6, line 222.

In addition, I suggest reviewing the writing style of the results, which, although correct, is repetitive.

Author Response

Dear,

I would like to express our thanks for your tremendous efforts in reviewing our paper. We have modified the manuscript accordingly, and the detailed corrections are listed below point by point:

Comment #1) I wonder if, for the sake of brevity, the authors could eliminate or summarize some of the descriptions included in the Results section for the determinations made. In some cases they are well-known methods in Natural Products studies (for example total phenols or total flavonoids).

Thank you very much for your suggestion. I have eliminated some of the descriptions included in the Results section for the determinations of total phenols or total flavonoids.

Comment #2) A few typos are detected in the manuscript, for example on page 3, line 132 and on page 6, line 222.

Thank you very much for your suggestion. I have corrected the typos in the manuscript.

Comment #3) In addition, I suggest reviewing the writing style of the results, which, although correct, is repetitive.

Thank you very much for your suggestion. I have made some corrections to the description of the results.

Sincerely Yours,

Chi-Rei Wu

Department of Chinese Pharmaceutical Sciences and Chinese Medicine Resources

China Medical University

crw@mail.cmu.edu.tw

Reviewer 2 Report

Comments and Suggestions for Authors

 Exploring the Neuroprotective Potential of Desmodium Species: Insights into Radical Scavenging Capacity and Mechanisms Against 6-OHDA-Induced Parkinsonism

The title suggests that the authors use a proven Parkinsonism experimental model. However, studies have shown that the effects of the extracts on their radical-scavenging and antioxidant properties in model chemical and cellular systems have been investigated. Yes, a full screening has been done in this aspect, with strong evidence confirmed by the content of polyphenols, flavonoids, and phenylpropanoids in the extracts.

Studies conducted in cellular systems, however, are not enough evidence to accurately predict whether these extracts can serve to control Parkinsonism. These precise cellular models that the authors have worked on are used to predict possible potential opportunities for controlling neurodegenerative conditions, which Parkinson's disease is likely to fall into.

I consider the discussion on this issue presented in this article to be not sufficiently convincing and partly manipulative.

What does the expression ... prevalent Desmodium plants mean?

Comments on the Quality of English Language

 Moderate editing of the English language is required. 

Author Response

Dear,

I would like to express our thanks for your tremendous efforts in reviewing our paper. We have modified the manuscript accordingly, and the detailed corrections are listed below point by point:

Comment #1) Studies conducted in cellular systems, however, are not enough evidence to accurately predict whether these extracts can serve to control Parkinsonism. These precise cellular models that the authors have worked on are used to predict possible potential opportunities for controlling neurodegenerative conditions, which Parkinson's disease is likely to fall into.

Thank you very much for your suggestion. I have changed Manuscript title from “6-OHDA-Induced Parkinsonism” to “6-OHDA-Induced Neurotoxicity”.

Comment #2) I consider the discussion on this issue presented in this article to be not sufficiently convincing and partly manipulative.

Thank you very much for your suggestion. This manuscript is based on the theories from previous relevant literature, followed by the collection of field research materials (Desmodium species). The results were obtained through rigorous research design and experimental procedures. Finally, a discussion is provided comparing the findings of this study with other related literature. Our results have both similarities and differences compared to other relevant literature. We have conducted a substantial discussion and drawn conclusions based on our research findings without overstepping the evidence.

Comment #3) Moderate editing of the English language is required.

Thank you very much for your suggestion. Grammatical and writing style errors in the original version have been corrected by MDPI English editing services.

Sincerely Yours,

Chi-Rei Wu

Department of Chinese Pharmaceutical Sciences and Chinese Medicine Resources

China Medical University

crw@mail.cmu.edu.tw

Reviewer 3 Report

Comments and Suggestions for Authors

In the present study, the antioxidant phytoconstituents and the radical scavenging capacities of these Desmodium species were investigated, and their effects on the activity of monoamine oxidase (NAO) and the autooxidation of 6-hydroxydopamine (6-OHDA) were also examined. Then, the neuroprotective effects of D. pulchellum on 6-OHDA-induced damage in SH-SY5Y cells. Consequently, D. pulchellum species were demonstrated to show the evident antioxidant and radical scavenging activities, thus suggesting that D. pulchellum would probably be able to cause the neuroprotective effects against 6-OHDA-induced Parkinsonism through radical scavenging activities of its antioxidant phytoconstituents and the restoration of intracellular antioxidant activities.

The section of “Abstract” seemed well-written, and it could explain the background and the objective of this study. The section of “Introduction” was easy to read, and uncomplicated to understand. The background and the rationalization were explained simply and clearly.

In the section of “Results”, the analyses and measurements were briefly explained by a simple description prior to the description of each experimental results, which was considered to help the readers to understand the meaning of presented results.

This work was logically designed , and carried out without any fatal defect. The manuscript seemed well-written. English was sound without any remarkable strangeness and oddness. But, some parts of the results would probably be better to move to the section of Discussion. Anyway, the manuscript was considered to be possibly good and reasonable, and also easy to read and understand.

Author Response

Dear,

I would like to express our thanks for your tremendous efforts in reviewing our paper. We have modified the manuscript accordingly, and the detailed corrections are listed below point by point:

Comment #1) But, some parts of the results would probably be better to move to the section of Discussion. Anyway, the manuscript was considered to be possibly good and reasonable, and also easy to read and understand.

Thank you very much for your suggestion. I have rewritten some parts of the results and discussion.

Sincerely Yours,

Chi-Rei Wu

Department of Chinese Pharmaceutical Sciences and Chinese Medicine Resources

China Medical University

crw@mail.cmu.edu.tw

Round 2

Reviewer 1 Report

Comments and Suggestions for Authors

The manuscript has been subtantially improved compared to v1.

However, there are still some writing aspects that can be improved:

- italicize the terms in vivo, in vitro

-check the similarity of some paragraphs in section 2.Results. See the lines highlighted in yellow in the attached document. As I expressed in my previous review, the style of presenting the results is repetitive.

-review the information about the collection of the plants, is it "mid- and low-altitude areas", or "low-altitude areas" (see lines highlighted in green in the attachment)

-review the wording on the lines highlighted in orange for inconsistences. Also, in lines 237, 286, 311, and 333 should be an IC50 value of...

Comments on the Quality of English Language

Language is adequate. Style can yet be improved.

Author Response

Dear,

I would like to express our thanks for your tremendous efforts in reviewing our paper. We have modified the manuscript accordingly, and the detailed corrections are listed below point by point:

Comment #1) italicize the terms in vivo, in vitro

Response #1: Thank you very much for your suggestion. I have collected it.

Comment #2) check the similarity of some paragraphs in section 2.Results. See the lines highlighted in yellow in the attached document. As I expressed in my previous review, the style of presenting the results is repetitive.

Response #2: Thank you very much for your suggestion. I have rewritten them.

Comment #3) review the information about the collection of the plants, is it "mid- and low-altitude areas", or "low-altitude areas" (see lines highlighted in green in the attachment)

Response #3: Thank you very much for your suggestion. The collection of the plants should be low-altitude areas. I have collected it.

Comment #4) review the wording on the lines highlighted in orange for inconsistences. Also, in lines 237, 286, 311, and 333 should be an IC50 value of...

Response #4: Thank you very much for your suggestion. I have collected it.

Sincerely Yours,

Chi-Rei Wu

Department of Chinese Pharmaceutical Sciences and Chinese Medicine Resources

China Medical University

crw@mail.cmu.edu.tw

Reviewer 2 Report

Comments and Suggestions for Authors

The authors have followed the reviewer's recommendations and made the recommended corrections.

Author Response

Dear,

I would like to express our thanks for your tremendous efforts in reviewing our paper. We have modified the manuscript accordingly, and the detailed corrections are listed below point by point:

Comment #1) The authors have followed the reviewer's recommendations and made the recommended corrections.

Response #1: Thank you very much for your suggestion.

Sincerely Yours,

Chi-Rei Wu

Department of Chinese Pharmaceutical Sciences and Chinese Medicine Resources

China Medical University

crw@mail.cmu.edu.tw
